# Natural 2′,4-Dihydroxy-4′,6′-dimethoxy Chalcone Isolated from *Chromolaena tacotana* Inhibits Breast Cancer Cell Growth through Autophagy and Mitochondrial Apoptosis

**DOI:** 10.3390/plants13050570

**Published:** 2024-02-20

**Authors:** Gina Mendez-Callejas, Marco Piñeros-Avila, Crispin A. Celis, Ruben Torrenegra, Anderson Espinosa-Benitez, Roberto Pestana-Nobles, Juvenal Yosa-Reyes

**Affiliations:** 1Grupo de Investigaciones Biomédicas y de Genética Humana Aplicada (GIBGA), Laboratorio de Biología Celular y Molecular, Facultad de Ciencias de la Salud, Universidad de Ciencias Aplicadas y Ambientales (U.D.C.A.), Calle 222 #55-37, Bogotá 111166, Colombia; marpineros@udca.edu.co (M.P.-A.); andeespinosa@udca.edu.co (A.E.-B.); 2Grupo de Investigación en Fitoquímica (GIFUJ), Departamento de Química, Facultad de Ciencias, Pontificia Universidad Javeriana, Cra. 7 #40-62, Bogotá 111321, Colombia; crispin.celis@javeriana.edu.co; 3Grupo de Investigación en Productos Naturales de la U.D.C.A. (PRONAUDCA), Laboratorio de Productos Naturales, Universidad de Ciencias Aplicadas y Ambientales (U.D.C.A.), Calle 222 #55-37, Bogotá 111166, Colombia; rtorrenegra@udca.edu.co (R.T.); 4Grupo de Investigación en Ciencias Exactas, Física y Naturales Aplicadas, Laboratorio de Simulación Molecular y Bioinformática, Facultad de Ciencias Básicas y Biomédicas, Universidad Simón Bolívar, Carrera 59 #59-65, Barranquilla 080002, Colombia; roberto.pestana@unisimon.edu.co

**Keywords:** *Chromolaena tacotana*, chalcone, breast cancer, intrinsic apoptosis, autophagy

## Abstract

Breast cancer (BC) is one of the most common cancers among women. Effective treatment requires precise tailoring to the genetic makeup of the cancer for improved efficacy. Numerous research studies have concentrated on natural compounds and their anti-breast cancer properties to improve the existing treatment options. *Chromolaena tacotana* (Klatt) R.M. King and H. Rob (*Ch*. *tacotana)* is a notable source of bioactive hydroxy-methylated flavonoids. However, the specific anti-BC mechanisms of these flavonoids, particularly those present in the plant’s inflorescences, remain partly undefined. This study focuses on assessing a chalcone derivative extracted from *Ch. tacotana* inflorescences for its potential to concurrently activate regulated autophagy and intrinsic apoptosis in luminal A and triple-negative BC cells. We determined the chemical composition of the chalcone using ultraviolet (UV) and nuclear magnetic resonance (NMR) spectroscopy. Its selective cytotoxicity against BC cell lines was assessed using the MTT assay. Flow cytometry and Western blot analysis were employed to examine the modulation of proteins governing autophagy and the intrinsic apoptosis pathway. Additionally, in silico simulations were conducted to predict interactions between chalcone and various anti-apoptotic proteins, including the mTOR protein. Chalcone was identified as 2′,4-dihydroxy-4′,6′-dimethoxy-chalcone (DDC). This compound demonstrated a selective inhibition of BC cell proliferation and triggered autophagy and intrinsic apoptosis. It induced cell cycle arrest in the G0/G1 phase and altered mitochondrial outer membrane potential (∆ψm). The study detected the activation of autophagic LC3-II and mitochondrial pro-apoptotic proteins in both BC cell lines. The regulation of Bcl-XL and Bcl-2 proteins varied according to the BC subtype, yet they showed promising molecular interactions with DDC. Among the examined pro-survival proteins, mTOR and Mcl-1 exhibited the most favorable binding energies and were downregulated in BC cell lines. Further research is needed to fully understand the molecular dynamics involved in the activation and interaction of autophagy and apoptosis pathways in cancer cells in response to potential anticancer agents, like the hydroxy-methylated flavonoids from *Ch. tacotana*.

## 1. Introduction

Breast cancer (BC) remains a significant public health challenge worldwide, despite advances in early detection. It comprises various molecular subtypes, many of which exhibit resistance to standard therapies [1]. Triple-negative breast cancer (TNBC) is a difficult type of breast cancer to treat and is associated with a poorer prognosis. This difficulty arises from its lack of estrogen receptors (ER), progesterone receptors (PR), and human epidermal growth factor receptor 2 (HER-2+) [1]. Other ER-positive breast cancer types include HER-2+, luminal A, and luminal B subtypes, with luminal A being the most prevalent. Luminal B breast cancer, while also ER-positive, is characterized by an overexpression of the *HER-2* gene [2].

Flavonoids are polyphenolic compounds found in nature that offer health benefits due to their biological properties, including antioxidant, antimicrobial, anti-inflammatory, and anticarcinogenic effects [3]. Chalcones are a subclass of flavonoids with great diversity in pharmacological activities. Structurally, they consist of two aromatic rings attached by three carbon units having α- and β-unsaturated ketone (1,3-diaryl-2-propen-1-one). The skeleton is an intermediate structure used in the biosynthesis of flavonoids [4]. Natural and synthetic chalcones have emerged as important molecules with various therapeutic properties as anticancer effects [5], including cytotoxicity for a wide range of cancer types, such as leukemia, hepatoma, colorectal cancer, stomach cancer, prostate cancer, epidermoid carcinoma, and breast cancer, through a mechanism than involve apoptosis induction by death receptor activation or Bcl-2 family protein regulation. The radical structures of chalcones have been associated with their anticancer properties. These structures include O-substitution of hydroxyl and methoxy groups, O-glycosylated and O-dimers chalcones, prenylated substitution (prenyl, geranyl, furano, and pyrano chalcones) with others, and the formation of Diels-Alder adducts (isoprene, monoterpene, coumarins, and other classes of flavonoids) [6].

Certain naturally occurring chalcones have shown effectiveness against breast cancer cells. Butein, for instance, influences estrogen metabolism by inhibiting the aromatase enzyme, with an IC_50_ value of 3.75 μM [7]. Isoliquiritigenin, a polyhydroxy chalcone derived from Glycyrrhiza glabra roots, has demonstrated both antitumorigenic activity and estrogen receptor (ER) α-dependent growth promotion in breast cancer cells, additionally inducing apoptosis [8,9].

Xanthohumol, a prenylated chalcone isolated from the female hop plant Humulus lupulus, exhibited in vitro antiproliferative effects and induced apoptosis in MDA-MB-231 breast cancer cells. This is achieved through the downregulation of Bcl-2 and matrix metalloproteinases (MMP-2 and MMP-9) at a concentration of 6.7 μM, which consequently activates Bax, Caspase-3, and Caspase-9 [10,11]. Isobavachalcone, sourced from Psoralea corylifolia L, enhances sensitivity to E2-induced paclitaxel resistance by down-regulating CD44 expression in ER-positive breast cancer cells [12]. Additionally, a prenylated chalcone named Xanthoangelol B, isolated from Ashitaba (Angelica keiskei Koidzumi, Apiaceae), induces apoptotic cell death through the activation of caspase-3, via a mechanism that does not involve the Bax/Bcl-2 signal transduction pathway [13].

Flavokavains (FKs) are a class of hydroxychalcones identified in various plants, including *Piper methysticum*, *P. triangularis* var. pallida, and *Didymocarpus corchorifolia* [14]. This group includes 2′-hydroxy 4,4′,6′-trimethoxychalcone (FKA), 6′-hydroxy-2′,4′-dimethoxychalcone (FKB), and 2′,4-dihydroxy-4′,6′-dimethoxy-chalcone (FKC). These compounds have demonstrated anticancer properties against MCF-7 and MDA-MB-231 breast cancer cell lines, inducing apoptosis and causing G2/M-phase cell cycle arrest. They also influence the processes of cell migration and invasion [15,16,17]. Specifically, FKB has been observed to induce cell death through p21-mediated cell cycle arrest and p38 activation in HeLa cervical cancer cells. Interestingly, its mechanism does not involve oxidative stress-induced apoptosis, as indicated by the concurrent activation of antioxidant-related pathways and iron sequestration pathways following the activation of ER-resident stress proteins [18]. FKC, in particular, has been studied for its potent inhibitory effects on colon cancer cell proliferation. It induces cell cycle arrest and promotes apoptosis, which is associated with endoplasmic reticulum stress and the regulation of MAPK and Akt signaling pathways [19]. Recent studies have shown that breast cancer cells treated with synthetic FKC exhibit suppressed colony formation, G2/M phase cell cycle arrest, apoptosis, and DNA damage in a dose-dependent manner [17].

The *Chromolaena* genus is distributed across temperate regions of Africa, Latin America, Southern Asia, and Australia [20]. Traditionally in ethnomedicine, *Chromolaena* species have been employed to treat a variety of ailments, including malaria, nasal congestion, inflammation, eye disorders, asthma, cough, flu, headache, colds, and, more recently, cancer [21,22]. Over 190 compounds have been isolated and identified from 27 species within this genus, encompassing a diverse array of chemical classes, such as flavonoids, alkaloids, triterpenoids, diterpenoids, sesquiterpenoids, steroids, fatty acids, coumarins, and chalcones [23]. 

Approximately nine chalcones have been isolated from the *Chromolaena* genus. These include pinocembrin chalcone (2′,4′,6′-Trihydroxychalcone) from *Chromolaena chasea* and several compounds from *Chromolaena odorata*: 2′,4-dihydroxy-4′,5′,6′-trimethoxy chalcone (ChDora), 2′-hydroxy-3′,4,4′,5′,6′-pentamethoxychalcone (Odoratin), 2′-hydroxy-4,4′,5′,6′-tetramethoxychalcone, 4,6′-dihydroxy-2′,3′,4′-trimethoxychalcone, 6′-hydroxy-4,2′,3′,4′-tetramethoxychalcone, 2′,4-dihydroxy-3′,4′,6′-trimethoxychalcone, 2′-hydroxy-3′,4,4′,6′-tetramethoxychalcone, and 2-hydroxy-4,′5,′6,′4,5-pentamethoxychalcone [23,24,25]. Odoratin has been observed to decrease cell viability in Cal51, HeLa, and MCF-7 breast cancer cells in a dose-dependent manner [23].

*Chromolaena tacotana* (Klatt) R. M. King and H. Rob (*Ch. tacotana*) is an endemic plant in Colombia, a source of flavonoids with antiproliferative properties [26] that depends on the position of the hydroxyl groups in the molecule’s structure [27]. Recently identified flavonoids, such as 3′,4′-dihydroxy-5,7-dimethoxy-flavanone and 2′,3,4-trihydroxy-4′,6′-dimethoxychalcone, exhibit significant antiproliferative activity against cancer cells, particularly TNBC [28,29]. Chalcotanina induces autophagy by affecting the structural conformation of the mTOR protein and apoptosis through the activation of caspases 3/7. Furthermore, it affects mitochondrial membrane potential and downregulates anti-apoptotic Bcl-2 members, thus activating the intrinsic pathway [29].

## 2. Results

The dichloromethane extract, derived from the inflorescences of the studied plant, was found to be rich in flavonoids. The specific chalcone investigated in this research was isolated from this flavonoid-rich extract. To elucidate the chemical structure of this compound, a combination of analytical techniques was employed. Ultraviolet (UV) spectroscopy data were collected, using various displacement reagents, including AcONa (sodium acetate), MeONa (sodium methoxide), AlCl_3_/HCl (aluminum chloride/hydrochloric acid), and H_3_BO_3_ (boric acid). Additionally, the structure was further characterized through both proton nuclear magnetic resonance (^1^H NMR) and carbon-13 nuclear magnetic resonance (^13^C NMR) spectroscopy. High-resolution electrospray ionization mass spectrometry (HR-ESI-MS) experiments also contributed to the comprehensive structural determination of this compound. These analytical methods collectively provided a detailed understanding of the molecular structure of the chalcone isolated from the dichloromethane extract.

### 2.1. Structural Analysis

The chalcone was obtained as an orange crystalline solid, soluble in acetone, and with a melting point of 196 °C. Its Rf value was 0.55 (silica gel, CHCl_3_:MeOH 9.8:0.2). It has a very faint yellow spot, almost white. It fluoresces yellow when viewed under UV light at λ = 366 nm and turns brown when exposed to NH_3_ vapors.

The ^1^H NMR (proton nuclear magnetic resonance) spectrum of the chalcone demonstrated 10 distinct signals, corresponding to 14 hydrogen atoms. This included two signals indicative of vinyl hydrogens, characteristic of unsaturated compounds like chalcones, and two signals for methoxy group hydrogens, suggesting methoxy substituents within the molecule. The pattern of meta-coupling signals in ring A pointed to a tetra-substitution configuration. In ring B, two symmetrical signals for four protons in meta-disposition suggested a double substitution. Additionally, the spectrum showed two signals for hydroxyl-type hydrogens, one in each ring, specifically at 8.96 ppm (ring B) and 14.41 ppm (ring A), confirming the molecule’s total hydrogen count as 16.

In the ^13^C NMR APT (Attached Proton Test) spectrum, 15 carbon signals were detected, accounting for 17 carbon atoms in the molecule. These signals included three for quaternary carbons, six for C-H carbons, two for O-CH_3_ carbons from methoxy groups, and four for C-O carbons. A significant chemical shift was observed at δ = 192.47 ppm, indicative of the carbonyl carbon common in flavonoids. The methoxy groups were further substantiated by signals at δ = 55.17 ppm and δ = 55.60 ppm in the negative phase. Additionally, intense signals at δ = 130.50 ppm and δ = 115.93 ppm in the negative phase suggested two pairs of methine-type carbons symmetrically positioned in ring B. These NMR analyses provide a detailed insight into the molecular structure of the chalcone, revealing its complex array of functional groups and substitution patterns.

The UV-Visible (UV-Vis) spectroscopy analysis of chalcone, particularly with the displacement reagents AcONa and H_3_BO_3_, provided further insights into its structure. The analysis indicated a likely hydroxyl substitution at the C-4 position in ring B. However, the possibility of a hydroxyl substituent at the C-4′ and C-2′ positions was ruled out, although the substitution of other radicals at these positions remained a possibility, as detailed on the right side of Table 1. Additionally, it was observed that the compound does not undergo hydrolysis in acidic conditions.

The exact mass of chalcone was calculated as 300.09588 ± 3.89 ppm. This calculation was based on data from electrospray ionization (ESI) mass spectrometry in both the negative and positive ion modes. In the negative ion mode, the [M − H]^−^ ion was measured at 299.08, and in the positive ion mode, the [M + H]^+^ ion was at 301.10. These results support the determination of the molecular formula as C_17_H_16_O_5_, which has a calculated mass of 300.09. This information, alongside the chemical structure representation provided in Figure 1 and additional details in Appendix A, allow for a comprehensive understanding of the chalcone’s molecular structure.

### 2.2. DDC Induces Selective Cytotoxicity in BC Cells

The study investigated the impact of DDC on the growth of two breast cancer (BC) cell lines, MCF-7 and MDA-MB-231, as well as on non-tumorigenic mammary gland cells, MCF-12F. This assessment was carried out using the MTT assay, a standard method for evaluating cell viability, with each well containing 10,000 cells. The results indicate that DDC reduced cell viability in a concentration-dependent manner, as illustrated in Figure 2A–C. A crucial aspect of the findings was the determination of the half-maximal inhibitory concentration (IC_50_) values, which measure the effectiveness of a substance in inhibiting a biological function. For the breast cancer cell lines MCF-7 and MDA-MB-231, the IC_50_ values were 52.5 µM and 66.4 µM, respectively. In contrast, the IC_50_ for the non-tumor MCF-12F cells was significantly higher at 232.8 µM, as shown in Figure 2D.

These IC_50_ values highlighted a marked disparity in cytotoxicity between the cancerous and non-cancerous cells, indicating a significant degree of selectivity (*p* < 0.001) of DDC towards the breast cancer cells. The selectivity index (SI), a ratio indicating the relative safety of the compound, was 4.4 for MCF-7 and 3.5 for MDA-MB-231 when compared to the SI values obtained for the positive controls, paclitaxel (PTX) and resveratrol (RV), as depicted in Figure 2E. This selectivity is important as it suggests that DDC might be effective in targeting breast cancer cells while having a lesser effect on normal cells, a desirable characteristic in cancer therapeutics.

### 2.3. Cell Cycle Progression in Response to the Natural Chalcone

DDC, identified as a naturally occurring chalcone, demonstrated a notable impact on the cell cycle of breast cancer cells. Specifically, it induced a blockade in the G1 phase in a significant majority of the cells within the studied breast cancer lines, MCF-7 and MDA-MB-231. This G1 phase arrest was observed in 65.7% of MCF-7 cells and 62.5% of MDA-MB-231 cells. In addition to this predominant G1 phase arrest, a smaller fraction of cells experienced cell cycle arrest in the G2/M phase, accounting for 24.1% in MCF-7 and 28.7% in MDA-MB-231 breast cancer cell lines.

These results were then compared to the effects of known chemotherapeutic agents, doxorubicin and combretastatin A-4, which were used as positive controls in the study. Doxorubicin is recognized for its ability to induce cell cycle arrest in the G1 phase, while combretastatin A-4 is known for its G2/M phase arrest capabilities. The findings with DDC show a closer resemblance to the cell cycle effects of doxorubicin rather than combretastatin A-4. The detailed outcomes of this comparison, including the specific phases of cell cycle arrest induced by each agent, are illustrated in Figure 3.

This ability of DDC to induce a G1 phase blockade in breast cancer cells highlights its potential as an anticancer agent, particularly considering its efficacy in targeting specific phases of the cell cycle, which is a crucial aspect of cancer treatment strategies.

### 2.4. Chalcone Targets Autophagy in BC Cells

The study aimed to assess the induction of autophagy in breast cancer (BC) cells exposed to DDC by detecting the LC3-II protein, a marker of autophagy. This protein localizes within autophagosomes, which are key structures in the autophagy process. Flow cytometry was utilized to detect LC3-II in chalcone-treated BC cells at 48 h, and the findings were corroborated by Western blot analysis at both 24 and 48 h.

The results reveal an increase in the LC3-II signal following DDC treatment in BC cells. Notably, the intensity of this signal and the proportion of LC3-II-expressing cells were higher in MCF-7 cells treated with DDC, surpassing even the levels detected in cells treated with resveratrol, which was used as a positive control, as depicted in Figure 4. Quantitatively, the autophagy levels, measured by the ratio of LC3-II-expressing cells in the treated cells compared to the negative controls, were found to increase by 7.7-fold in MCF-7 and 6.3-fold in MDA-MB-231 cells (Figure 4B).

In addition, Western blot and immunofluorescence assays supported the findings of increased LC3-II protein expression in DDC-treated BC cells, which was more pronounced in MCF-7 cells. The study also analyzed the phosphorylation of mTOR, and the protein level of Beclin-1, two key regulator proteins in autophagy. The levels of phosphorylated mTOR reduced over time during the treatment, while Beclin-1 increased specially in MCF-7 cells. Depending on the cell type, the effect of p62, a specific cargo receptor for autophagy, varied. While p62 expression decreased significantly in MDA-MB-231 cells in a time-dependent manner, there was not a significant decrease in MCF-7 cells up to 48 h after chalcone treatment, as shown in Figure 4C.

These findings underscore the potential of DDC to induce autophagy in BC cells, with variations in response between different cell lines. The upregulation of LC3-II and the modulation of autophagy-related proteins, like mTOR, Beclin-1, and p62, highlight the complex interplay of cellular mechanisms influenced by the DDC treatment.

### 2.5. BC Genotype-Dependent Variations in Apoptosis Triggering by DDC

To determine the apoptotic effect of DDC on breast cancer (BC) cells, flow cytometry was employed for analyses at 24 and 48 h post-treatment, using the IC_50_ concentration of DDC. Paclitaxel (PTX) served as the positive control, while untreated cells were the negative control. The study utilized two assays: the detection of Annexin-V binding to externalized phosphatidylserine (PS) on the cell membrane, a marker of early apoptosis, along with the DNA dye 7-AAD to differentiate between early and late apoptosis stages (Figure 5A,B). Furthermore, flow cytometric analysis was used to assess the presence of active caspases 3 and/or 7, with or without 7-AAD, confirming the induction of apoptosis in BC cells by DDC (Figure 5C,D).

However, a notable resistance to complete the apoptosis process was observed in caspase 3-deficient MCF-7 cells [2], which predominantly remained in the early phase of apoptosis even after 48 h, as indicated by both Annexin-V and effector caspase 7 detection. Conversely, a significant proportion of MDA-MB-231 cells advanced to late apoptosis at 48 h. This led to a higher overall rate of apoptosis in TNBC cells.

Further analysis through Western blotting confirmed a higher level of active caspase 3, and caspase 7 was particularly evident in MDA-MB-231 cells, while caspase 7 apoptosis-dependent activity was confirmed in caspase 3-deficient MCF-7 cells. Additionally, the study evaluated XIAP, a caspase inhibitor protein, revealing a significant reduction in its expression in MDA-MB-231 cells post-treatment, in contrast to MCF-7 cells, where no significant change was noted (Figure 5E,F).

These findings highlight the variable response of different BC cell lines to DDC-induced apoptosis, particularly noting the stronger apoptotic effect in TNBC cells compared to others, and underscoring the potential therapeutic relevance of DDC for breast cancer treatment.

### 2.6. TNBC and Luminal A BC Cells Differ in the Pathway Leading to Intrinsic Apoptosis in Response to DDC

Alterations in the mitochondrial outer membrane potential (∆ψm) were observed in breast cancer (BC) cells after treatment with the natural compound DDC, as depicted in Figure 6. According to flow cytometric analysis, both BC cell lines, namely MDA-MB-231 and MCF-7, exhibited depolarization similar to that induced by the positive control, valinomycin. Notably, a higher proportion of MDA-MB-231 cells demonstrated a loss of membrane potential compared to the MCF-7 cells, with approximately 70% and 58% affected, respectively, as shown in Figure 6B.

A Western blot analysis was conducted to assess the protein levels involved in the intrinsic apoptosis pathway. Densitometric data were normalized based on GAPDH expression as a loading control. In both breast cancer (BC) cell lines treated with DDC, there was an increase in the expression of the pro-apoptotic proteins p53, Bax, and Bim. However, the elevation in p53 levels was not statistically significant. Regarding the anti-apoptotic Bcl-2 family proteins, notable differences were observed. In MCF-7 cells, a significant reduction in Bcl-2 and Mcl-1 protein levels was noted, while Bcl-XL levels increased at 48 h post-treatment. In contrast, in MDA-MB-231 cells, Bcl-2 levels initially decreased up to 24 h, but returned to baseline levels at 48 h post-treatment, as shown in Figure 7. In the same cell line, both Bcl-XL and Mcl-1 exhibited a time-dependent decrease in protein levels.

Flow cytometry was employed to detect the phosphorylation of the Bcl-2 protein at serine 70 (S70pBcl-2), which represents its active form, as well as its non-phosphorylated, inactive state. In untreated breast cancer (BC) cells, the S70pBcl-2 form was predominantly observed. However, the treatment with chalcone resulted in a significant dephosphorylation of this protein, leading to its inactivation. Specifically, in MCF-7 cells, dephosphorylation reached up to 97%, and in MDA-MB-231 cells, up to 78% after 48 h of treatment, as demonstrated in Figure 7D,E. This effect mirrored the response observed with the positive control, paclitaxel (PTX).

### 2.7. In Silico Analysis

#### 2.7.1. Docking and Molecular Dynamics (MDs) Simulation between DDC and Pro-Survival Proteins

An *in silico* analysis, incorporating both molecular docking and molecular dynamics (MDs) simulations, was utilized to investigate the interactions between DDC ligand and the five key proteins that play a role in negatively regulating apoptosis: Bcl-2, Bcl-XL, Mcl-1, mTOR, and XIAP, as referenced in other studies [30,31]. The initial phase involved molecular docking studies, which aimed to delineate possible binding modes and calculate the binding affinity of DDC to each of the selected proteins. This was followed by MDs simulations, which provided insights into the stability of the resulting ligand–protein complexes over time. To gain a more nuanced understanding of the binding dynamics, the Molecular Mechanics Poisson–Boltzmann Surface Area (MM-PBSA) method was subsequently applied, offering a refined computation of the binding free energy associated with these interactions.

In the molecular docking studies, AutoDock Vina was employed as the computational tool, as cited in references [32,33]. The results, illustrated in Figure 8, reveal that all the proteins under study exhibited similar binding energies when interacting with the DDC ligand. It is noteworthy that, among these proteins, XIAP displayed the lowest binding energy, indicating a potentially stronger interaction with the DDC ligand compared to the others.

The analysis of the root-mean-square deviation (RMSD) values, as depicted in Figure 9A–E, revealed that all proteins demonstrated stable interactions with the DDC ligand throughout the molecular dynamics simulation. Notably, the mTOR protein exhibited a particularly enhanced stability in its interaction with the DDC ligand, especially after the 40-nanosecond (ns) mark, compared to its stability when not bound to any ligand. This finding is further supported by the analysis of the root-mean-square fluctuation (RMSF) values, shown in Figure 9F–J. The lower fluctuation levels observed in the systems containing the DDC ligand suggest that the presence of the ligand contributes to the overall stabilization of the system.

Figure 10 illustrates the mTOR protein as a dimer, with the DDC ligand positioned at its interface. The root-mean-square deviation (RMSD) and root-mean-square fluctuation (RMSF) values collectively underscore the stabilizing effect of the DDC ligand on the mTOR dimer structure. Additionally, the proteins Bcl-2, Bcl-XL, Mcl-1, and XIAP, when complexed with the DDC ligand, show minimal variations in both RMSD and RMSF, indicating stable interactions. A notable observation is the high mobility of the loop regions at either the N-terminus or C-terminus of these proteins, highlighting their inherent flexibility.

#### 2.7.2. Analysis of the Interaction between DDC and Pro-Survival Proteins 

To decipher the interactions occurring throughout the molecular dynamics (MDs) simulations, the Prolif Python library was utilized for a detailed analysis, as referenced in [34]. The analysis encompassed the entire trajectory of each simulation. Residues that were engaged in interactions for more than 30% of the simulation’s duration were specifically considered. Additionally, particular types of interactions, such as hydrophobic interactions and van der Waals (vdW) contacts, were accounted for if they occurred for over 10% of the trajectory. Figure 11 showcases both a bar plot and a two-dimensional plot that together illustrate the range and frequency of these various interactions.

Predominantly, two types of interactions were discerned between the DDC ligand and the target proteins: hydrophobic interactions and vdW contacts. Hydrophobic interactions are notably critical for the stabilization of protein–ligand complexes, as they play a significant role in the systems observed, as supported by references [35,36]. Additionally, π-stacking interactions, though less prevalent, were also observed among the proteins.

For each protein, distinct interactions with the DDC ligand were identified. Bcl-2 exhibited hydrogen bond acceptor (HBAcceptor) interactions at residues GLN25 and SER24. Bcl-XL demonstrated π-cation interactions at ARG99, along with hydrogen bond acceptor interactions at the same residue and hydrogen bond donor (HBDonor) interactions at GLY89 and GLY98. MCL-1 showed hydrogen bond acceptor interactions at THR96, MET80, and HIE54. mTOR revealed hydrogen bond acceptor interactions at SER140, and hydrogen bond donor interactions at multiple residues, including TRP206, PHE58, ASP207, GLN66, PHE213, THR203, and VAL59. Lastly, XIAP displayed hydrogen bond acceptor interactions at TYR76, ASN1, and GLY58 and hydrogen bond donor interactions at TYR76 and GLY58.

#### 2.7.3. Binding Free Energy Analysis Using MM-PBSA

To evaluate the stability of the protein–ligand interactions, the binding free energy of each complex was calculated utilizing the Molecular Mechanics Poisson–Boltzmann Surface Area (MM-PBSA) method. This analysis encompassed the entire duration of the simulation trajectories, as illustrated in Figure 12. The results indicate that the proteins mTOR, Mcl-1, and Bcl-XL show comparable binding free energy values. In contrast, the Bcl-2 protein exhibits the highest binding free energy among the studied proteins.

## 3. Discussion

Approximately 79 flavonoids, comprising 9 chalcones, have been isolated and identified from the genus *Chromolaena*, as noted in reference [23]. This research culminated in the isolation and identification of a specific chalcone derived from the inflorescences of *Chromolaena tacotana*. The experimental molecular weight of this chalcone was calculated to be 300.09588 unified atomic mass units (u.m.a), corresponding to the molecular formula C_17_H_16_O_5_.

The ^1^H NMR spectrum of chalcone exhibits characteristic signals. Methoxyl groups are evident in high fields, with chemical shifts at δ = 3.88 and 4.02 ppm. Additionally, signals around δ = 7.78 and 7.91 ppm, each integrating for one hydrogen, correspond to vinyl hydrogens coupled to α and β carbons, a feature typical of chalcones. At δ = 6.10 and 6.13 ppm, a coupling constant (J_HH) of 2.4 Hz is observed, indicative of meta-coupling in ring A, suggesting a meta-tetra-substituted pattern. In ring B, signals at δ = 6.94 and 7.64 ppm (doublet of doublets, J = 2.4 Hz) correlate to four protons in meta positions, implying a double substitution in a symmetrical arrangement, as summarized in Table 1.

The ^13^C NMR APT spectrum reveals 15 signals for 17 carbons. A signal at δ = 192.42 ppm identifies the carbonyl carbon, a hallmark of the flavonoid structure. The signals at δ = 55.62 ppm and 55.17 ppm in the negative phase are attributed to methoxyl groups. This molecule presents five characteristic carbon signals in the negative phase linked to a proton (CH), while the remaining are quaternary carbons. The signals at δ = 168.26, 166.38, 162.79, and 159.83 ppm are noteworthy, each being linked to an oxygen atom.

The COSY H-H spectrum elucidates the vicinal coupling between the C-α and C-β protons, confirming the presence of vinylic protons in the structure. It also reveals the ortho-positioning of two methines (CH) in ring B. The signals at δ = 13.050 ppm and δ = 115.93 ppm are pronounced in the negative phase, indicative of two pairs of methine-type carbons in a symmetrical chemical environment, located in ring B. This symmetry and the observed chemical shifts suggest a para-substitution pattern in the ring, as detailed in Table 1.

Based on the connectivity established between various hydrogens and their corresponding carbons, as determined through HMQC and HMBC experiments, the compound under study was identified as 2′,4-dihydroxy-4′,6′-dimethoxychalcone, as illustrated in Figure 1. The NMR data for this chalcone were closely aligned with the spectral assignments of a novel chalcone synthesized earlier, where fifteen peaks were observed in the ^13^C NMR spectrum. Specifically, the peak at 193.2 ppm was assigned to a carbonyl group. The peaks with double intensities at 55.6, 106.5, and 161.2 ppm corresponded to 3′-OMe/5′-OMe, C-2′/C-6′, and C-3′/C-5′, respectively. Additionally, the peak at 55.9 ppm was attributed to the 2-OMe′ group, as documented in reference [37].

The mass of chalcone, as calculated experimentally, aligns with the structure elucidated through NMR analysis, exhibiting variations of less than 5 ppm. Due to their acidic nature, chalcones are prone to easy deprotonation. In the mass spectral analysis, this particular molecule demonstrated a loss of 30 *m*/*z*, which is consistent with the sequential elimination of two CH_3_ groups (each with *m*/*z* 15). Moreover, a prominent peak at *m*/*z* 119 was observed, likely resulting from the cleavage between the carbonyl group and the alpha carbon, including ring B. Another notable cleavage pathway, indicated by an *m*/*z* of 180, involves the initial deprotonation of ring B leading to an intermediate of *m*/*z* 164, followed by a subsequent loss of CO, yielding a fragment of *m*/*z* 136.

This behavior parallels findings reported by Zhang [38], who characterized the structures of similar compounds. Zhang noted common neutral losses in methoxy-containing chalcones, including H_2_, H_2_O, CO, CO_2_, and methyl radicals. He further elucidated that specific neutral losses followed stepwise pathways, such as the loss of *m*/*z* 44, which can be attributed to the combination of CH_3_ and HCO, or CH_4_ and CO, along with the elimination of CO_2_.

Numerous studies have been conducted on the cytotoxic activity of both natural and synthetic chalcones. A significant limitation in many of these studies is the absence of data on the selectivity index (SI), which is crucial for evaluating the therapeutic potential of these compounds. SI is defined as the ratio of the toxic concentration to the effective bioactive concentration in normal cells [39]. The current study demonstrated that the natural chalcone DDC, derived from *Chromolaena tacotana*, exhibited pronounced cytotoxicity against breast cancer (BC) cells. This aligns with previous findings on the cytotoxic properties of synthetic flavokawains (FKCs), where MCF-7 cells displayed a greater sensitivity compared to MDA-MB-231 cells, as indicated in Figure 2. Notably, the IC_50_ values for these cell lines were found to be 52.5 µM and 66.4 µM, respectively, as shown in Figure 2D. These variations may be attributable to differences in the cell seeding density and the methodology used for absorbance measurement at 570 nm, a recommended wavelength for assessing formazan formation [40].

In addition, the cytotoxicity of DDC was evaluated on non-tumorous MCF-12 mammary gland cells to determine the selectivity index (SI). Our findings reveal an SI value exceeding 3.5 (Figure 2E), suggesting a low adverse impact on the surrounding healthy cells. According to Weerapreeyakul et al. [41], an SI value above 3.0 is indicative of a compound’s potential as an effective anticancer agent. These results underscore the importance of the SI in assessing the therapeutic viability of such compounds, thereby warranting further investigation.

During the treatment with DDC, a notable alteration in the normal cell cycle progression was observed in both breast cancer (BC) cell lines, leading to cell cycle arrest in the G0/G1 phase, as depicted in Figure 4. This observation contrasts with the results of a previous study [42], which reported that flavokawains (FKCs) induce cell cycle arrest in the G2/M phase. The experiments conducted in our study utilized synchronized cells, ensuring consistency in cell cycle stages across samples. The results obtained with DDC are in alignment with the effects of doxorubicin, a widely used chemotherapeutic agent for breast cancer treatment. Doxorubicin is known to cause cell cycle arrest in the G0/G1 phase in these BC cells, as corroborated by findings in the literature [43,44].

Previous research indicates that natural chalcones, including butein, xanthohumol, isoliquiritigenin, chalcotanina, and flavokawains, may exhibit cytotoxic effects, particularly inducing apoptosis in breast cancer (BC) cells [17,28,43,45]. However, prior to this study, there was no evidence demonstrating the simultaneous induction of both apoptosis and autophagy in BC cells by flavokawains (FKCs). Our research unveiled that the treatment with DDC triggers not only apoptosis but also a significant autophagic response, as indicated by the detection of the LC3-II protein. Notably, the proportion of cells undergoing autophagy was higher in MCF-7 cells in comparison to MDA-MB-231 cells. In contrast, triple-negative breast cancer (TNBC) cells, represented by MDA-MB-231, exhibited a more pronounced induction of apoptosis. This is substantiated by the increased levels of pro-apoptotic proteins and the downregulation of XIAP and other anti-apoptotic proteins, as confirmed by Western blot analysis.

A range of flavonoids have been identified as inducers of autophagy in breast cancer cells by inhibiting the Akt/mammalian target of the rapamycin (mTOR) pathway [46]. In our study, we observed that DDC not only inhibited mTOR but also elevated LC3-II protein levels in both breast cancer (BC) cell lines, indicating the induction of autophagy. The p62 protein exhibited a persistent expression in MCF-7 cells, but not in triple-negative breast cancer (TNBC) cells, indicating a notable difference. Autophagy activation is through selective and non-selective pathways. In selective autophagy, p62 serves as a transport protein, transferring ubiquitinated substrates to the autophagosome for degradation, which is subsequently degraded gradually [47]. It is feasible that p62 experiences swift degradation in TNBC cells, but not in MCF-7 cells, where autophagy seems to be more active based on our findings, as explained in Figure 4.

The induction of apoptosis by chalcones, particularly through the activation of caspase-3, is a well-documented phenomenon [28,29,30,31,32,33,34,35,36,37,38,39,40,41,42,43,44,45,46,47,48,49]. In our study, the data suggested that DDC preferentially activated caspase-3 over caspase-7 in MDA-MB-231 cells, as illustrated in Figure 5. Once activated, cleaved caspase-3 translocated to the nucleus, where it plays a crucial role in the cleavage of DNA and other substrates, thereby participating in the proteolytic events characteristic of apoptosis [50,51]. On the other hand, caspase-7 is primarily involved in the detachment of apoptotic cells from the adjacent extracellular matrix (ECM) [52], as occurred in deficient-caspase 3 MCF-7 cells. Additionally, our findings indicate a reduction in the expression of the XIAP protein following DDC treatment. This reduction could explain the activation of effector caspases, possibly through the disruption of their interaction with the BIR2 domain located in the N-terminal region of XIAP [53], thereby facilitating the apoptotic process, as depicted in Figure 5.

Anticancer agents are known to modify mitochondrial permeability, often resulting in depolarization, the loss of mitochondrial membrane potential (∆ψm), and the subsequent release of apoptotic factors [54]. Our data demonstrated that treatment with DDC initiates the mitochondrial pathway of apoptosis in breast cancer (BC) cells. This was evidenced by the reduction in ∆ψm (Figure 6), accompanied by increased levels of the mitochondrial pro-apoptotic proteins Bax and Bim. Conversely, the levels of the anti-apoptotic proteins Mcl-1, Bcl-2, and Bcl-XL decreased following DDC treatment. However, there was a notable tendency for Bcl-XL in MCF-7 and Bcl-2 in MDA-MB-231 cells to restore their protein levels after 48 h of treatment, which suggests that tumor cells might activate defense mechanisms to counteract the stress induced by DDC [55]. This response appears to be dependent on specific anti-apoptotic Bcl-2 family members, varying between different types of BC. Notably, the downregulation of Mcl-1 is consistent in both luminal A and triple-negative BC (TNBC).

Additionally, the proteins Mcl-1, Bcl-2, and Bcl-XL are known to negatively regulate autophagy. Their inhibition, as observed in our study, leads to the release of Beclin-1, which forms a core complex essential for autophagosome nucleation [56]. These findings collectively support the simultaneous induction of autophagy and intrinsic apoptosis by DDC in BC cells.

Bcl-2 phosphorylation at serine 70 (S70pBcl-2) plays a critical role in forming anti-apoptotic heterodimers with Bax or Bak. These heterodimers contribute to the enhancement in the outer mitochondrial membrane’s permeability, thereby inhibiting apoptosis [57]. Our study’s findings reveal a significant decrease in Bcl-2 phosphorylation following DDC treatment, with reductions of 97% in luminal A and 78% in triple-negative breast cancer (TNBC) cells. This substantial decrease in phosphorylation disrupts the anti-apoptotic Bak/Bcl-2 and Bax/Bcl-2 complexes, potentially facilitating the initiation of intrinsic apoptosis and possibly autophagy.

In silico studies, encompassing molecular docking, molecular dynamics (MDs) simulations, and Molecular Mechanics Poisson–Boltzmann Surface Area (MM-PBSA) calculations, provided substantial insights into the interaction dynamics between the DDC ligand and a set of pro-survival proteins, including Bcl-2, Mcl-1, Bcl-XL, XIAP, and mTOR, as analyzed in vitro. The molecular docking studies initially offered preliminary insights into the potential binding affinities and interaction modes, which were further refined through MDs simulations (Figure 8, Figure 9, Figure 10, Figure 11 and Figure 12). Notably, each protein displayed stable interactions with DDC, as indicated by the root-mean-square deviation (RMSD) and root-mean-square fluctuation (RMSF) analyses (Figure 9). However, the data suggested that DDC binding did not introduce significant perturbations or induce notable conformational changes within these proteins (Figure 10).

The interactions between the proteins and the DDC ligand primarily involved hydrophobic and vdW contacts, essential for stabilizing the respective complexes. These interactions are likely a combination of hydrogen bonding, hydrophobic contacts, and ionic interactions, all contributing to the stabilization of DDC within the protein binding pockets. Moreover, the solvation effects, as calculated by the PBSA component of MM-PBSA, also play a crucial role in this stabilization process, highlighting the importance of solvent molecules in mediating these interactions [58].

While the Bcl-2 anti-apoptotic proteins demonstrated favorable interactions with DDC, they did not achieve the binding energy levels observed with mTOR. This discrepancy could be due to either suboptimal interactions or potential repulsive forces, emphasizing the complex nature of protein–ligand interactions. Additionally, factors like ligand strain upon binding and entropic contributions might influence the binding energies.

It is important to note that the binding free energy is a cumulative measure of various energetic components, each with its significance in the binding process (Figure 11). Although the Bcl-2 protein showed a high binding free energy, it also exhibited a complex network of interactions, including hydrogen bonds, hydrophobic, and vdW contacts, likely leading to its inactivation by dephosphorylation in both MCF-7 and MDA-MB-231 cells (Figure 7D,E). Furthermore, the solvation effects, as part of the MM-PBSA calculation, emphasized the pivotal role of solvent molecules in these interactions, significantly contributing to the overall stability of the protein–ligand complexes.

## 4. Materials and Methods

### 4.1. Chalcone Extraction and Isolation

The plant material used in this study was collected from La Vega, Cundinamarca, Colombia, at an altitude of 1250 m above sea level (m.a.s.l.). A specimen of the plant was deposited for identification at the Herbarium of Pontificia Universidad Javeriana, Bogotá, Colombia. It was identified as *Chromolaena tacotana* (Klatt) R.M. King and H. Rob, with the voucher number HPUJ30170.

The inflorescences, dried at 60 °C to less than 8% humidity, were ground in a RTE-648 mill (Tecnal, Sao Paulo, Brazil) and sieved through a 60-mesh screen (0.25 mm) (EMS-08, Electrolab, Mumbai, India). A total of 490.17 g of the resulting powder underwent Soxhlet extraction using petroleum ether (Sigma-Aldrich, St Louis, MO, USA). This was followed by dichloromethane (CH_2_Cl_2_) extraction (Merck Group, Darmstadt, Germany) to remove fats and chlorophylls. The solvent was then evaporated under vacuum at 45 °C using a R-300 rotary vacuum evaporator (BÜCHI Labortechnik AG, Flawil, Switzerland) at 60 rpm. The residue, termed the DCI fraction, weighed 30.82 g and was measured using an Explorer balance (Ohaus, NJ, USA).

The DCI fraction was flocculated with a methanol (MeOH) and water mixture (1:1) (Merck Group, Darmstadt, Germany). Subsequent liquid–liquid extraction with CH_2_Cl_2_ yielded the DC-II fraction. A total of 7.8563 g of the DC-II fraction was then subjected to column chromatography (CC) using silica gel (63–200 μm) (Silicagel 60, Merck Group, Darmstadt, Germany). The elution process involved a gradient of petrol:ethyl acetate (AcOEt) with ratios of 8:2, 7:3, and 6:4, followed by ethanol (EtOH) (Merck Group, Darmstadt, Germany), resulting in five separate fractions. Fraction 3 underwent further CC using a chloroform (CHCl_3_) and methanol (MeOH) mixture (9.8:0.2), yielding 162.2 µg of a single compound. This compound was subsequently purified and crystallized through successive washings with acetone (Merck Group, Darmstadt, Germany).

### 4.2. Structural Identification

^1^H and ^13^C NMR spectra were acquired using an Avance Bruker 300 spectrophotometer (Bruker Biospin, Mannheim, Germany), operating at the frequencies of 300 MHz for hydrogen and 75 MHz for carbon. The spectra were recorded in acetone-d6 solvent (Merck Group, Darmstadt, Germany). Mass spectral data were obtained using ultra-high-pressure liquid chromatography (UPLC) coupled to a quadrupole time-of-flight (QTOF) mass spectrometer detector, specifically a Shimadzu LCMS-9030 (Kyoto, Japan), via direct injection. The positions of O-CH_3_ and OH groups were determined using UV-VIS spectroscopy within the range from 230 to 500 nm, conducted on a Jenway 6405 UV-VIS spectrophotometer (Staffordshire, UK). This analysis utilized displacement reagents, including AlCl_3_, HCl, AcONa, MeONa, and H_3_BO_3_ (Merck Group, Darmstadt, Germany).

### 4.3. Breast Cell Lines and Culture Conditions

The MDA-MB-231 (HTB-26) and MCF-7 breast cancer cell lines (American Type Culture Collection (ATCC), Manassas, VA, USA) were cultured in RPMI 1640 medium (Lonza, Basel, Switzerland). The cancer cell media were supplemented with 10% heat-inactivated fetal bovine serum (Biowest, Nuaillé, France) and 1.0% penicillin/streptomycin (Lonza, Basel, Switzerland). Conversely, the normal epithelial MCF-12F (CRL-10783) mammary gland cells (ATCC, Manassas, VA, USA) were grown in DMEM/F-12 medium (Sigma-Aldrich, St Louis, MO, USA). The DMEM/F-12 medium was enriched with 7.0% fetal horse serum, 10 µg/mL human insulin, 20 ng/mL epidermal growth factor, 500 ng/mL hydrocortisone, 100 mg/mL cholera toxin, and 1.0% penicillin/streptomycin, all obtained from Sigma-Aldrich, St Louis, MO, USA.

### 4.4. Cytotoxic Activity and Selectivity of 2′,4-Dihydroxy-4′,6′-Dimethoxychalcone

The cytotoxic activity of DDC was assessed using the MTT (Sigma-Aldrich, St. Louis, MO, USA) reduction assay, specifically measuring the inhibitory concentration (IC_50_) that reduces cell viability by 50%. A total of 10,000 cells were seeded in each well of 96-well plates and treated with various concentrations of DDC, ranging from 80 to 5 µg/mL. The cells treated with 0.5% DMSO (Thermo-Fisher Scientific, Waltham, MA, USA) served as the negative control. Following a 48 h incubation period at 37 °C and 5.0% CO_2_ in a Binder BD-115 incubator (Tuttlingen, Germany), 0.5 mg/mL MTT reagent was added to each well and incubated for an additional 4 h. The resulting Formazan crystals were dissolved in DMSO, and the absorbance was measured at 570 nm using a Bio-Rad microplate reader (Bio-Rad, Hercules, CA, USA). The IC_50_ values were estimated through nonlinear regression analysis using GraphPad Prism 8.0 software (La Jolla, CA, USA). The selectivity index (SI) was calculated as the ratio of the IC_50_ in normal cells to the IC_50_ in cancer cells [39,58].

### 4.5. Effects of 2′,4-Dihydroxy-4′,6′-dimethoxychalcone on Cell Cycle Progression in BC Cells

To evaluate the impact of DDC on the cell cycle distribution, specifically the G0/G1, S, and G2/M phases, flow cytometry analysis was conducted using the Muse™ Cell Analyzer (Luminex Corporation, Austin, TX, USA). A total of 100,000 cells per well were seeded in a 24-well plate and, after attachment, were treated with DDC at the IC_50_ concentration for 24 h at 37 °C and 5.0% CO_2_. Combretastatin A-4,Doxorubicin and Paclitaxel (Luminex Corporation, Austin, TX, USA) were used as positive controls for the G2/M [59] and G0/G1 cell cycle arrest [60], respectively. Synchronized, untreated cells served as the negative control.

Post-treatment, the cells underwent trypsinization and were resuspended in 1X PBS (Lonza, Basel, Switzerland). They were then gradually fixed in fresh, cold 70% ethanol and incubated at 4 °C for 24 h. The Muse^®^ Cell Cycle Kit (Luminex Corporation, Austin, TX, USA), which employs PI-based staining to quantify DNA content, was used to distinguish and measure the percentage of cells in each phase of the cell cycle (G0/G1, S, and G2/M), following the manufacturer’s instructions. Data acquisition was performed using the MUSE cell analyzer. During this process, values were generated and displayed on a histogram, with the distribution of cells being compared against the positive and negative control populations.

### 4.6. Susceptibility of BC Cells to Autophagy Induced by Natural Chalcone

#### 4.6.1. Detection of LC3-II using a Flow Cytometry Assay

Autophagy activation was analyzed by flow cytometry through the detection of LC3-II in DDC-treated cells.. For this purpose, 40,000 cells per well were seeded in a 48-well plate. Once attached, these cells were exposed to DDC at the IC_50_ concentration and incubated for 24 h at 37 °C and 5.0% CO_2_. As positive controls for autophagy, cells subjected to overnight starvation and resveratrol treatment were used [61], while cells treated with 0.5% DMSO (Thermo-Fisher Scientific, Waltham, MA, USA) served as the negative control.

Following the 24 h treatment period, the Muse^®^ Autophagy Kit (Luminex Corporation, Austin, TX, USA), which includes an anti-LC3-II Alexa Fluor™ 555 antibody, was utilized according to the manufacturer’s guidelines. Data acquisition was carried out using the MUSE cell analyzer (Luminex Corporation, Austin, TX, USA). During this process, the values were collected and displayed on a histogram graph. For each sample, the data were expressed as the median percentage of cells that tested either positive or negative for the LC3-II autophagic marker.

#### 4.6.2. Western Blot Analysis

To verify the induction of autophagy, a Western blot analysis was conducted using 40 µg of the total protein extracted from the breast cancer (BC) cells. The cells were lysed using Cell Lysis Buffer II (Thermo-Fisher Scientific, Waltham, MA, USA), following treatment with DDC for 24 and 48 h as well as with the positive control resveratrol (Rv). The proteins were separated on 7.5 or 12% SDS-PAGE gels and subsequently transferred onto polyvinylidene fluoride (PVDF) membranes (both from Thermo-Fisher Scientific, Waltham, MA, USA). The membranes were blocked for 1 h using 5.0% (*w*/*v*) BSA in TTBS and then incubated overnight at 4 °C with primary antibodies: anti-mTOR-Ser2448 (Proteintech, Rosemont, Illinois, USA), anti-Beclin-1 (D40C5) (Cell Signaling Technology, Denver, MA, USA), anti-p62, and anti-LC3-II (GeneTex, Irvine, CA, USA). Anti-GAPDH (GT239) (GeneTex, Irvine, CA, USA) was employed as the loading control.

Following primary antibody incubation, the membranes were incubated for 2 h at room temperature with either anti-mouse IgG or anti-rabbit IgG secondary antibodies (1:5000) (Merck Group, Darmstadt, Germany) [28]. Bands were visualized using the 1-Step™ TMB-Blotting Substrate Solution (Thermo-Fisher Scientific, Waltham, MA, USA), which generates a blue precipitate. Band density was normalized and semi-quantified using the ImageJ V 1.8.0 software.

#### 4.6.3. Morphological Analysis of Nuclei and LC3-II Expression by Epifluorescent Microscopy

BC cells were seeded in a 24-well plate. After adherence and growth to 60% confluence, the cells were treated with DDC and incubated at 37 °C in 5.0% CO_2_ for 24 h. DNA staining was performed with DAPI (Sigma-Aldrich, St Louis, MO, USA), and LC3-II protein was labeled with anti-LC3-II (GeneTex, Irvine, CA, USA) and Goat anti-Rabbit IgG (H+L) Cross-Adsorbed Secondary Antibody, Texas Red (Thermo-Fisher Scientific, Waltham, MA, USA). Images were captured using a MoticCamPro 282A and analyzed at 40× using the Motic Image Plus 2.0 software (Schertz, TX, USA).

### 4.7. Apoptosis in Response to 2′,4-Dihydroxy-4′,6′-dimethoxychalcone

#### 4.7.1. Apoptosis

Live cell detection and the differentiation between early or late apoptosis were conducted using the Muse^®^ Annexin V & Dead Cell Reagent Kit and the Muse^®^ Caspase-3/7 Kit (both from Luminex Corporation, Austin, TX, USA). For these assays, 40,000 breast cancer (BC) cells per well were seeded in a 24-well plate and treated with DDC at the IC_50_ concentration for 24 h. Paclitaxel (Luminex Corporation, Austin, TX, USA) was used as the positive control, while cells exposed to 0.5% molecular-grade DMSO (Thermo-Fisher Scientific, Waltham, MA, USA) served as the negative control.

Following treatment, both the medium and cells were collected by centrifugation, washed in 1X PBS, and then resuspended. The cells were subsequently stained with the reagent solutions, according to the manufacturer’s protocols provided in each Muse^®^ Kit (Luminex Corporation, Austin, TX, USA). Apoptotic cells were detected using the Guava^®^ Muse^®^ Cell Analyzer (Luminex Corporation, Austin, TX, USA), with the analysis being performed using the Muse^®^ software version 1.8.

#### 4.7.2. Western Blotting

Untreated breast cancer cells, after being treated with DDC for 24 and 48 h, and cells treated with paclitaxel (PTX) as the positive control, were processed following the method described in Section 4.6.2. For the analysis, these cells were examined using specific antibodies: anti-caspase-7 (GTX31704), anti-cleaved caspase-3 (Thermo Fisher Scientific, Waltham, MA, USA), and anti-XIAP (D2Z8W) (Cell Signaling Technology, Denver, MA, USA).

### 4.8. Analysis of the Intrinsic Pathway of Apoptosis in TNBC Cells

#### 4.8.1. Changes in the Outer Mitochondrial Membrane Potential (∆ψm)

To assess changes in the mitochondrial membrane potential (∆ψm) in response to DDC treatment, a flow cytometry analysis was performed. Twenty-five thousand cells were seeded in each well of a 48-well plate and incubated overnight at 37 °C with 5.0% CO_2_. The adherent cells were then treated with DDC at the IC_50_ concentration and 100 nM Valinomycin for 24 h. The Muse^®^ MitoPotential Kit (Luminex Corporation) was utilized to determine whether DDC induces depolarization of the mitochondrial membrane. Changes in ∆ψm in the triple-negative breast cancer (TNBC) cells were detected using the Guava^®^ Muse^®^ Cell Analyzer (Luminex Corporation) and analyzed using the Muse^®^ software version 1.8.

#### 4.8.2. Pro- and Anti-Apoptotic Protein Analysis Using Western Blot

To identify pro- and anti-apoptotic mitochondrial proteins in breast cancer (BC) cells, 40 µg of the total proteins extracted from the untreated cells and the cells treated with chalcone (DDC) for 24 and 48 h, as well as the cells treated with paclitaxel (PTX) as a positive control, were processed following the protocol outlined in Section 4.6.2. The analysis employed the following antibodies: anti-Bim-EL (GT1234), anti-Bcl-2 (N1N2) (GeneTex, Irvine, CA, USA), anti-Bax (D2E11), anti-Bcl-XL (54H6), and anti-Mcl-1 (D2W9E) (Cell Signaling Technology, Denver, MA, USA), to determine the levels of pro- and anti-apoptotic proteins.

#### 4.8.3. Phospho-Bcl-2 Protein Detection

To analyze the phosphorylation of the Bcl-2 protein at serine 70 (S70pBcl-2), flow cytometry was employed. Initially, two hundred thousand cells were seeded in each well of a 12-well plate and incubated overnight at 37 °C with 5.0% CO_2_. Post-incubation, the adherent cells were treated with chalcone (DDC) and paclitaxel (PTX) at their respective IC_50_ concentrations for 48 h. The assessment of Bcl-2 protein phosphorylation was conducted using the Muse™ Bcl-2 Activation Dual Detection Kit (Luminex Corporation), which includes two directly conjugated antibodies: anti-phospho-Bcl-2 (Ser70)-Alexa Fluor^®^555 for the active form and anti-Bcl-2-PECy5 for the inactive form of Bcl-2.

Following treatment, the cells were dissociated, harvested, and processed for fixation, permeabilization, and staining. Data acquisition and analysis were carried out using the Guava^®^ Muse^®^ Cell Analyzer (Luminex Corporation) and analyzed using the Muse® soft-ware version 1.8. The experiment was conducted in triplicate under identical conditions, and the results are presented as the mean of these replicates.

### 4.9. Statistical Analysis for the Biological Test

Data from the experiments were presented as the mean ± standard deviation (SD) of three biological replicates (n = 3). To determine the statistical significance of the differences between the control group and treatment groups, a two-way ANOVA test followed by Tukey’s multiple comparison test was employed. These statistical analyses were conducted using the GraphPad Prism software, version 8.0 (La Jolla, CA, USA). Statistical significance was established at *p*-values of <0.05 (**) and <0.001 (***), indicating significant differences between the means.

### 4.10. Molecular Docking and Molecular Dynamics

Molecular docking simulations were performed using AutoDock Vina version 1.2.0 (San Diego, CA, USA) [32,33], utilizing the Vina force field to predict and evaluate the binding affinities between the selected proteins and ligands. The search space for the docking was defined by a cubic box, centered on the ligands, with each side measuring 35 Å. A grid spacing of 0.375 Å was set as the default for these simulations.

The exhaustiveness parameter in the docking simulations was set to 50, a value chosen to ensure an efficient and thorough search for the global minimum within the designated search space. This setting strikes a balance between computational cost and the accuracy of the docking predictions. The docking pose with the lowest binding free energy was selected from these simulations to serve as the initial configuration for subsequent molecular dynamics simulations.

Protein structures were sourced from the Protein Data Bank (PDB, Piscataway, NJ, USA), with the respective PDB IDs for the proteins under study being: Mcl-1 (5FDR) [31], Bcl-XL (3ZLN), XIAP (5OQW), mTOR (4JSV), and Bcl-2 (6O0P). Structures that included an inhibitor bound to the protein were preferred. In cases where such structures were unavailable, the CastP server was utilized to predict and analyze protein cavities, pockets, or tunnels.

For the docking preparations, water molecules and other solvents were removed from the protein structures using the ChimeraX 1.7.1 software (San Francisco, CA, USA) [62]. Subsequently, the proteins were converted into the pdbqt format using the AutoDock Tools of the ADFR Suite [TO10b]. This step involved the addition of polar hydrogen atoms and the assignment of Gasteiger partial charges to the atoms, thereby generating the necessary pdbqt files required for the docking simulations.

The ligands were prepared for docking using the MEEKO 0.5.0 software (San Francisco, CA, USA), available at https://github.com/forlilab/Meeko (accessed on 12 April 2023). MEEKO’s functionalities include the addition of polar hydrogen atoms to the ligands (where necessary) and the computation of Gasteiger charges. This processing results in the generation of the corresponding pdbqt files, which are suitable for use in docking simulations.

#### 4.10.1. Molecular Dynamics Simulation Preparations

For the ligands, new charges were calculated to achieve a more accurate representation of the molecules. RESP charges were computed using Gaussian 16 (Wallingford, CT, USA) with the B3LYP/6-311g basis set [63]. The parameters and coordinate files for the ligands were generated using Antechamber, a component of AmberTools [64]. The systems comprising protein–ligand complexes were prepared using Leap, another tool of AmberTools. For these simulations, the ff14SB force field [65] was employed for proteins, while the GAFF force field [66] was used for ligands. Each system was then solvated with TIP3P water molecules [67], and Na^+^ or Cl^−^ ions were added to neutralize the system’s charge.

#### 4.10.2. Molecular Dynamics Configuration

To enhance the molecular-level description of the ligands, their charges were recalculated using the Restricted Electrostatic Potential (RESP) method. This recalibration was carried out using the Gaussian 16 software suite 16 (Wallingford, CT, USA) [63], employing the B3LYP functional and the 6-311g basis set. These settings were chosen to accurately represent the electron correlation and distribution within the ligands. The quantum mechanical calculations performed in Gaussian were pivotal for deriving the electrostatic potential, which was then used to extract the RESP charges.

After completing the quantum mechanical calculations, the Antechamber tool was employed to create parameter and coordinate files for the ligands. This tool was essential for converting the calculated RESP charges and molecular coordinates into a format compatible with the AMBER suite of programs (San Francisco, CA, USA), thereby facilitating their use in subsequent simulation processes.

The preparation of the protein–ligand complexes, or systems, was carried out using the Leap module of the AmberTools package [64]. Leap was instrumental in assigning force field parameters to the atoms in the system and generating the required input files for molecular dynamics simulations. The ff14SB force field [65] was applied to the protein components to provide accurate potential energy parameters for simulating protein dynamics. For the ligands, the General Amber Force Field (GAFF) [66] was chosen due to its versatility and effectiveness in representing a wide range of organic molecules.

Each system, consisting of a protein–ligand complex, was solvated in a periodic box of TIP3P water molecules [67]. The TIP3P model was selected for its reliable representation of water’s structural and dynamic properties, thereby offering a physiologically relevant simulation environment. To maintain the overall neutrality of the system, which is crucial for the stability and accuracy of the simulations, either sodium (Na^+^) or chloride (Cl^−^) ions were added to counteract the net charge of the protein–ligand complexes.

Molecular dynamics simulations were conducted using the AMBER18 simulation package (San Francisco, CA, USA) [64], following a carefully structured protocol. The initial phase of the simulation focused on the energy minimization of water molecules in the system. This step was executed for a maximum of 5000 cycles (specified by maxcyc = 5000), transitioning from steepest descent to conjugate gradient optimization after the first 1000 cycles (controlled by ncyc = 1000). During this phase, the SHAKE algorithm was disabled (ntf = 1) to permit the unrestrained movement of hydrogen atoms. Concurrently, the solute, comprising the protein–ligand complex, was stabilized with positional restraints, applying a force constant of 100 kcal/mol-Å^2^. This restraint was essential to maintain the protein structure during the water minimization process.

After the initial minimization of water molecules, the entire system was subjected to a comprehensive energy minimization. This step aimed to eliminate potential steric clashes and alleviate strained molecular conformations. The process entailed up to 100,000 cycles of minimization (set by maxcyc = 100,000), transitioning from steepest descent to conjugate gradient methods after the initial 1000 cycles (controlled by ncyc = 1000). During this stage, the SHAKE algorithm remained disabled (ntf = 1), allowing for the free movement of all atoms in the system.

Following energy minimization, the system underwent a heating phase, where its temperature was gradually increased to 300 K over 500 picoseconds (ps). During this phase, the protein–ligand complex (solute) was subject to positional restraints with a reduced force constant of 2 kcal/mol-Å^2^. This allowed the system to slowly adapt to the increasing temperature within an NVT ensemble (constant number of particles, volume, and temperature).

After the heating phase, the system was subjected to pressure equilibration for an additional 500 picoseconds (ps) within an NPT ensemble (constant number of particles, pressure, and temperature). This step allowed the system to adapt to the desired pressure conditions. Subsequently, the system underwent an equilibration phase for 1 nanosecond (ns), followed by a production run of 10 ns. Trajectories from the final 2 ns of the production run were extracted and utilized for Molecular Mechanics Poisson–Boltzmann Surface Area (MM-PBSA) calculations, in accordance with methodologies established in the previous literature.

#### 4.10.3. MMPBSA

The binding free energies of the protein–ligand complexes, following molecular dynamics simulations, were calculated using the MMPBSA.py module (San Francisco, CA, USA) of the AmberTools18 package [68]. The Molecular Mechanics Poisson–Boltzmann Surface Area (MM-PBSA) method is known for striking a balance between accuracy and computational efficiency in predicting binding free energies in molecular complexes.

For our analysis, the entire trajectory from the production phase of the molecular dynamics simulation was used. The input parameter (inp = 1) was set to sequentially read and process trajectory frames, enabling a detailed and continuous analysis of the system’s energy profile throughout the simulation. We opted not to use the radii optimization feature (radiopt = 0), maintaining a consistent set of radii derived from the force field parameters for both the simulation and subsequent analysis.

In our calculations, we chose to exclude the entropy contribution to the binding free energy. While entropy is an important factor in the binding process, its accurate computation is computationally intensive and does not always enhance the precision of binding free energy estimations. In some instances, including entropy can result in less accurate predictions [69,70]. Therefore, considering the balance between computational resources and accuracy, entropy was omitted from our binding free energy calculations in this study.

#### 4.10.4. RMSD, RMSF, and Rg

The analyses of the root-mean-square deviation (RMSD), root-mean-square fluctuation (RMSF), and radius of gyration (Rg) were conducted using the cpptraj module [71] from the AMBER20 software suite (San Francisco, CA, USA). These calculations provide crucial insights into the structural stability, flexibility, and compactness of the protein–ligand complexes over the course of the simulation.

For RMSD calculations, which assess the conformational stability, the first frame of the production phase served as the reference structure. The RMSD values were calculated for each frame against this reference, offering a temporal profile of structural deviations during the simulations.

RMSF analysis, conducted with the same reference frame, provides details on the flexibility and mobility of individual residues or atoms in the protein. This analysis highlights regions with significant conformational changes throughout the simulation.

The Rg was calculated to assess the compactness and overall dimensions of the proteins, effectively measuring their size and conformational changes over time, which is crucial for understanding protein folding and stability.

Additionally, the Prolif Python library [34] was employed for a detailed assessment of protein–ligand interactions throughout the simulation trajectory. This library allows for the analysis of these interactions frame-by-frame over the entire simulation, providing a comprehensive view of the dynamic interplay between the protein and the ligand. This interaction analysis is vital for understanding the ligand’s binding specificity and affinity towards the protein, offering key insights into their molecular interaction mechanisms.

## 5. Conclusions

This study suggests that 2′,4-dihydroxy-4′,6′-dimethoxychalcone, derived from *Chromolaena tacotana*, may possess therapeutic potential for breast cancer treatment, operating through the differential activation of autophagy and mitochondrial apoptosis in a cell-type-dependent manner. The selective cytotoxic effects on MCF-7 and MDA-MB-231 breast cancer (BC) cells were characterized by the simultaneous induction of autophagy and apoptosis, along with cell cycle arrest in the G0/G1 phase. Increased levels of LC3-II and inhibition of the mTOR protein, which directly and stably interacted with DDC, promoted autophagy in MCF-7 cells. Conversely, in triple-negative breast cancer (TNBC) cells, the emphasis was on the activation of mitochondrial apoptosis. This was marked by a decrease in the mitochondrial membrane potential and the consequent activation of pro-apoptotic proteins, such as Bax, Bim, and p53, following the deregulation of Mcl-1, Bcl-XL, Bcl-2, and XIAP, leading to caspase-3- and/or -7-mediated apoptosis. These findings support the anticancer potential of *Ch tacotana*.

## Figures and Tables

**Figure 1 plants-13-00570-f001:**
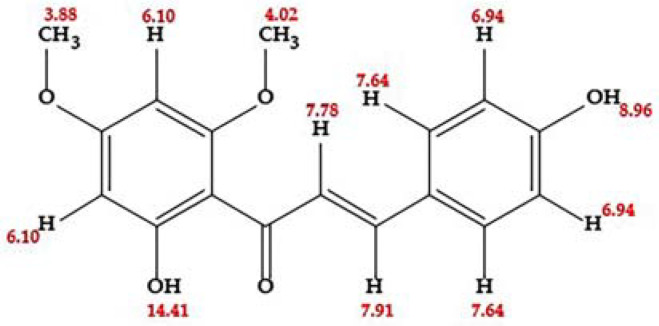
Structure of 2′,4-dihydroxy-4′,6′-dimethoxy-chalcone (DDC).

**Figure 2 plants-13-00570-f002:**
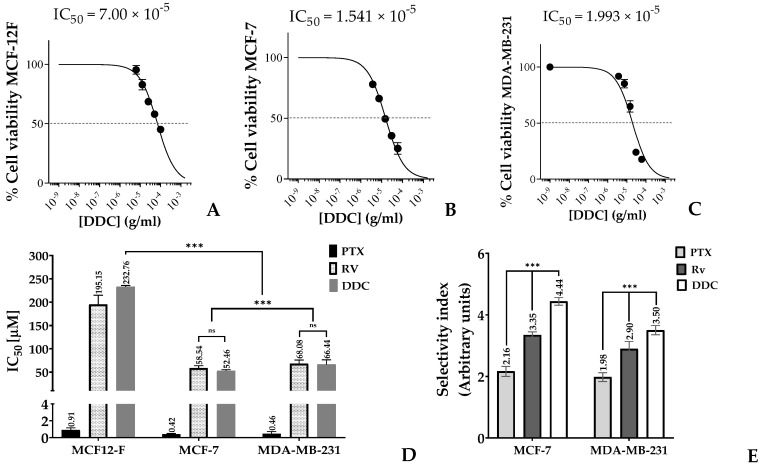
Selective inhibition of cell viability in response to DDC. (**A**–**C**) MCF12-F, MCF-7, and MDA-MB-231 percentages of cell viability in response to chalcone treatment at 48 h. (**D**) Half-maximal concentration (IC_50_) and (**E**) selectivity index of the BC cells, using the positive control for apoptosis (paclitaxel (PTX)) and for autophagy (resveratrol (RV)). Data from three independent experiments, each performed in triplicate, were obtained and data were analyzed using GraphPad Prism 8.0 (La Jolla, CA, USA). *p*-values indicate statistical significance (*** *p* < 0.001) or ns: not significant.

**Figure 3 plants-13-00570-f003:**
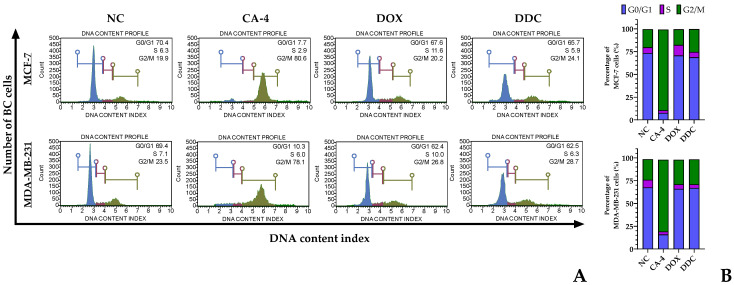
Flow cytometric analysis of the cell cycle in BC cells in response to DDC. (**A**) Histograms show the distribution of untreated cells or the negative control (NC) combretastatin A-4 (CA-4) and doxorubicin (DOX)-treated cells used as positive controls of the G2/M phase (in green) and the G0/G1 phase (in blue), and DDC-treated cells at IC_50_ for 24 h and stained with propidium iodide. The S phase is shown in violet. (**B**) Percentage of the total untreated and treated MCF-7 and MDA-MB-231 BC cells distributed in each phase of the cell cycle (G0/G1, S, and G2/M).

**Figure 4 plants-13-00570-f004:**
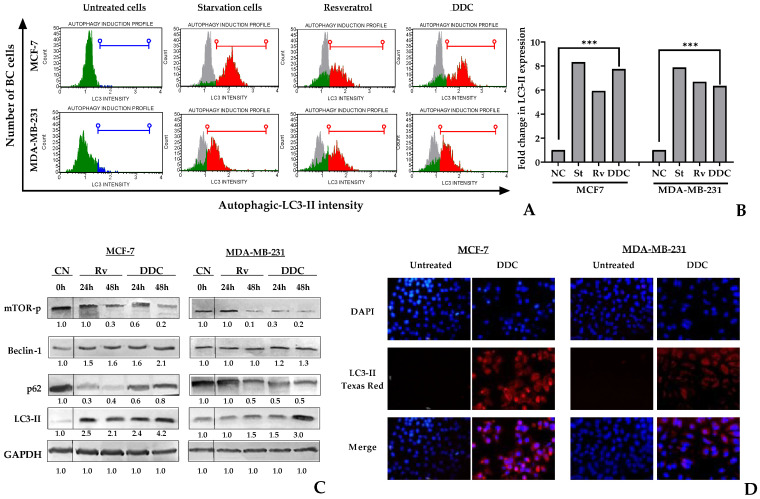
Induction of autophagy in BC cells in response to DDC. (**A**) Histogram plots representing negative and positive LC3-II detection in BC cells after DDC treatment in green and red, respectively. Untreated cells or the negative control (NC) are in grey; starving cells and resveratrol-treated cells were used as positive controls. (**B**) The fold change of LC3-II expression was calculated between the treated and untreated cells, providing a quantitative measure of the autophagy induced in response to treatment.. (**C**) Western blot of mTOR-Ser2448, Beclin-1, p62, and LC3-II proteins. Semiquantitative analysis was performed according to the relative densitometric units obtained by using ImageJ, later normalized against the negative controls. (**D**) Immunofluorescence microscopy images showing the expression of LC3-II in response to DDC (in red) and nuclei (in blue) at 24 h. *p*-values indicate statistical significance (*** *p*-value < 0.001).

**Figure 5 plants-13-00570-f005:**
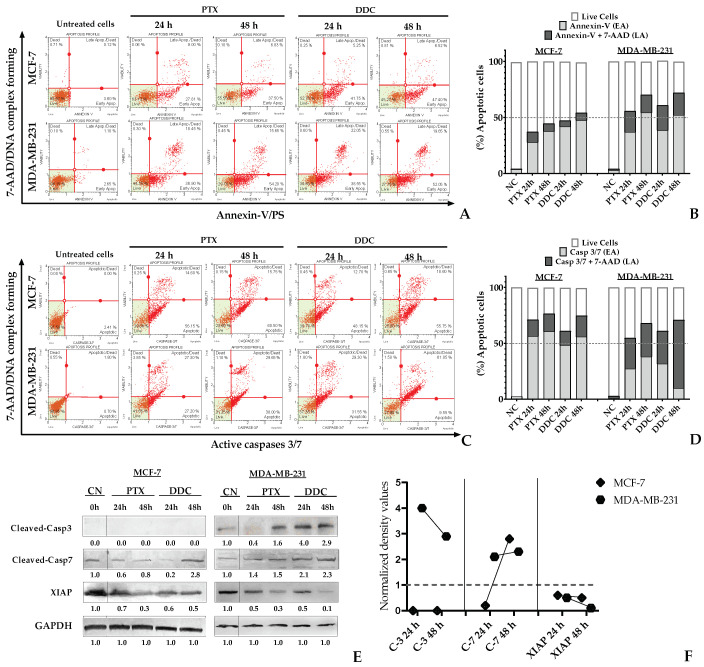
Apoptosis induction on MCF-7 and MDA-MB-231 cells in response to DDC at 24 h and 48 h. EA (early apoptosis); LA (late apoptosis); NC corresponds to untreated cells. PTX was used as the positive control. (**A**,**C**) The dot (in red) represents a single cell analyzed, and localized live cells are in the green area. (**A**) The upper panel corresponds to Annexin V/7-AAD detection on cells. (**B**) The bar graph indicates the percentage of cells in early (Annexin-V) or late (Annexin-V/7-AAD) apoptosis. (**C**) The bottom panel corresponds to the activation of caspases 3 and/or 7. (**D**) The bar graph indicates the percentage of cells in early or late apoptosis according to the activation of caspases. (**E**,**F**). The protein expression levels of cleaved caspase 3 (C-3) and caspase 7 (C-7) in DDC-treated BC cells were analyzed using Western blot. Normalized density values were calculated with respect to the negative control.. GAPDH was used as a loading control.

**Figure 6 plants-13-00570-f006:**
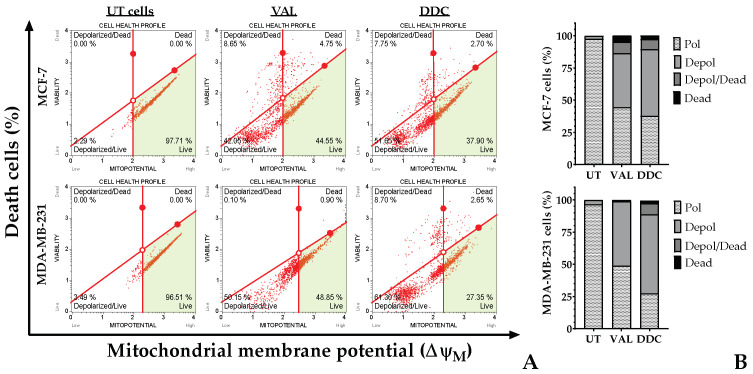
Flow cytometric analysis of mitochondrial membrane potential (∆ψm) in BC cells exposed to DDC at 24 h. (**A**) Red dot represents each analyzed BC cell. The green area represents the region of cells with high ∆ψm or live cells. Valinomycin at 100 nM served as a positive control (VAL), while untreated cells (UT) were used as the negative control. (**B**) Distribution of percentage of BC cells with a polarized or depolarized mitochondrial membrane.

**Figure 7 plants-13-00570-f007:**
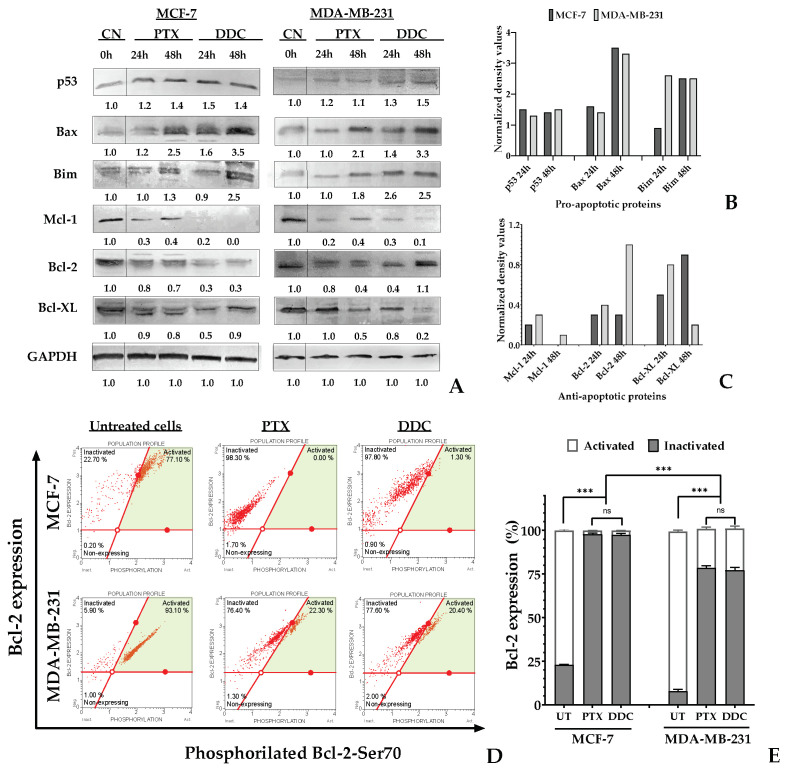
Intrinsic apoptosis induction in BC cells in response to DDC. (**A**) Western blot analysis of the pro-apoptotic p53, Bax, and Bim and the anti-apoptotic Mcl-1, Bcl-2, and Bcl-XL protein levels in BC cells treated with DDC and the positive control PTX for 24 and 48 h. (**B**,**C**) The bar graphs of normalized density values of pro- and anti-apoptotic protein levels in DDC-treated BC cells. (**D**) Flow cytometry analysis showing density plots of the status of the Bcl-2 protein in BC cells treated with DDC and PTX for 48 h. The X-axis represents the Bcl-2-Ser70 protein form, while the Y-axis represents the total expression of the Bcl2 protein. The dot (in red) represents a single cell analyzed, and the green region corresponds to the area of active Bcl-2 cells (**E**). The bar graph indicates the percentage of untreated (UT) and treated cells with active and inactive Bcl-2 protein expression. *p*-values indicate statistical significance (*** *p* < 0.001) or ns: not significant.

**Figure 8 plants-13-00570-f008:**
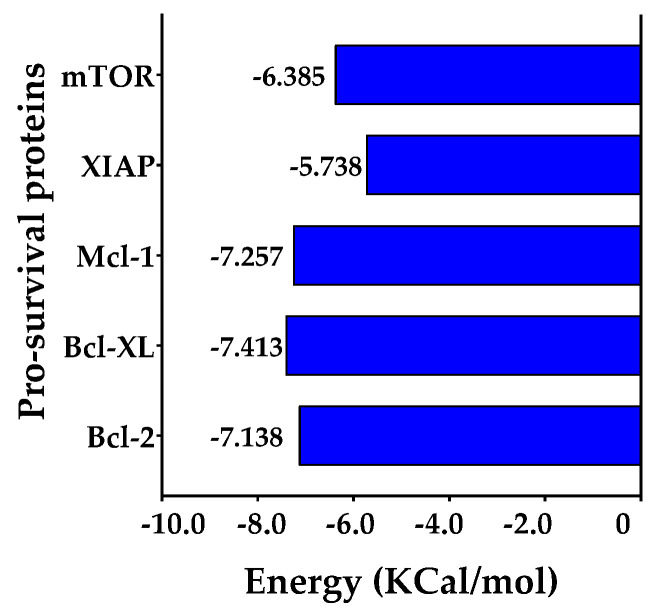
Docking results showing the binding affinities of the DDC ligand with each protein.

**Figure 9 plants-13-00570-f009:**
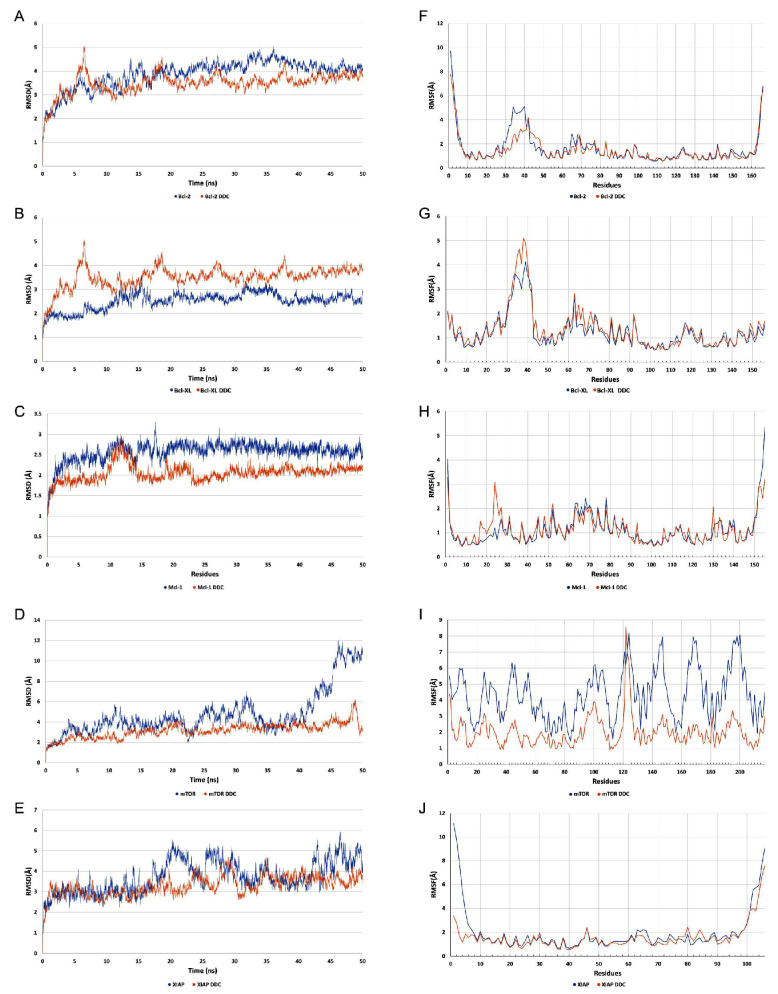
(**A**–**E**) display the root-mean-square deviation (RMSD) trajectories for several regulatory proteins interacting with DDC. Specifically, these include (**A**) Bcl-2, (**B**) Bcl-XL, (**C**) Mcl-1, (**D**) mTOR, and (**E**) XIAP. In these figures, the RMSD values for the standalone proteins are represented in blue, whereas the orange traces illustrate the RMSD of the protein–ligand complexes, indicating their interactions with DDC. (**F**–**J**) provide the root-mean-square fluctuation (RMSF) profiles for the same set of proteins: (**F**) Bcl-2, (**G**) Bcl-XL, (**H**) Mcl-1, (**I**) mTOR, and (**J**) XIAP. Here, the unliganded state of each protein is depicted in blue, while the orange traces signify their bound states with DDC.

**Figure 10 plants-13-00570-f010:**
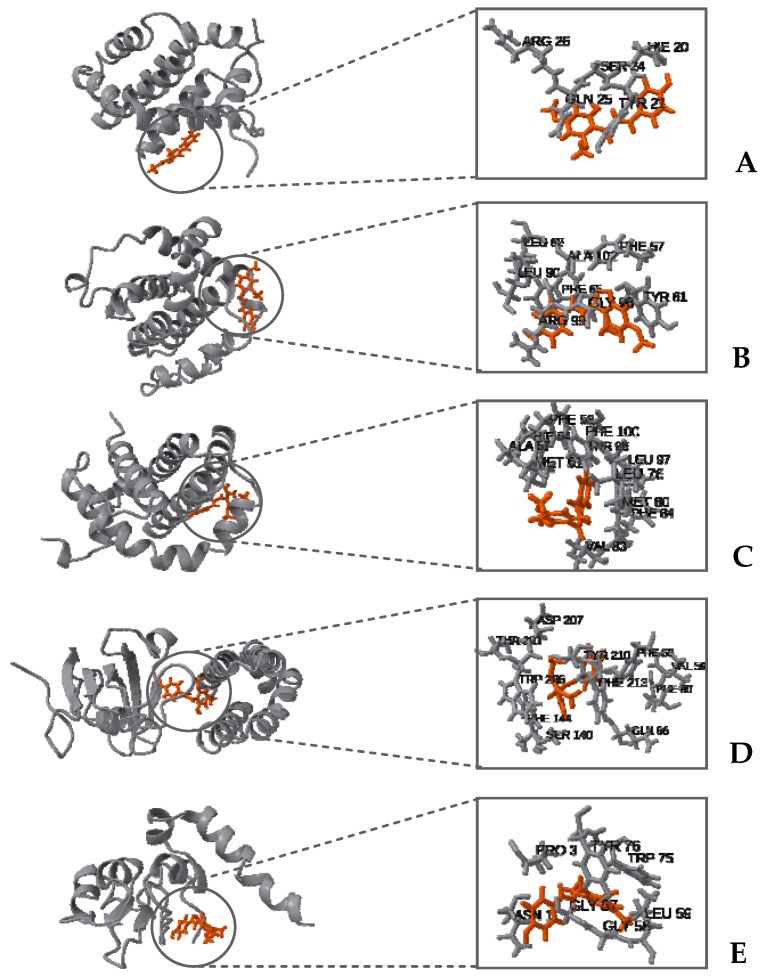
Comprehensive view of the pro-survival proteins complexed with the ligand DDC, highlighting the major interacting residues. Proteins are depicted in gray, with the DDC ligand represented in orange. (**A**) Bcl-2, (**B**) Bcl-XL, (**C**) Mcl-1, (**D**) mTOR, and (**E**) XIAP.

**Figure 11 plants-13-00570-f011:**
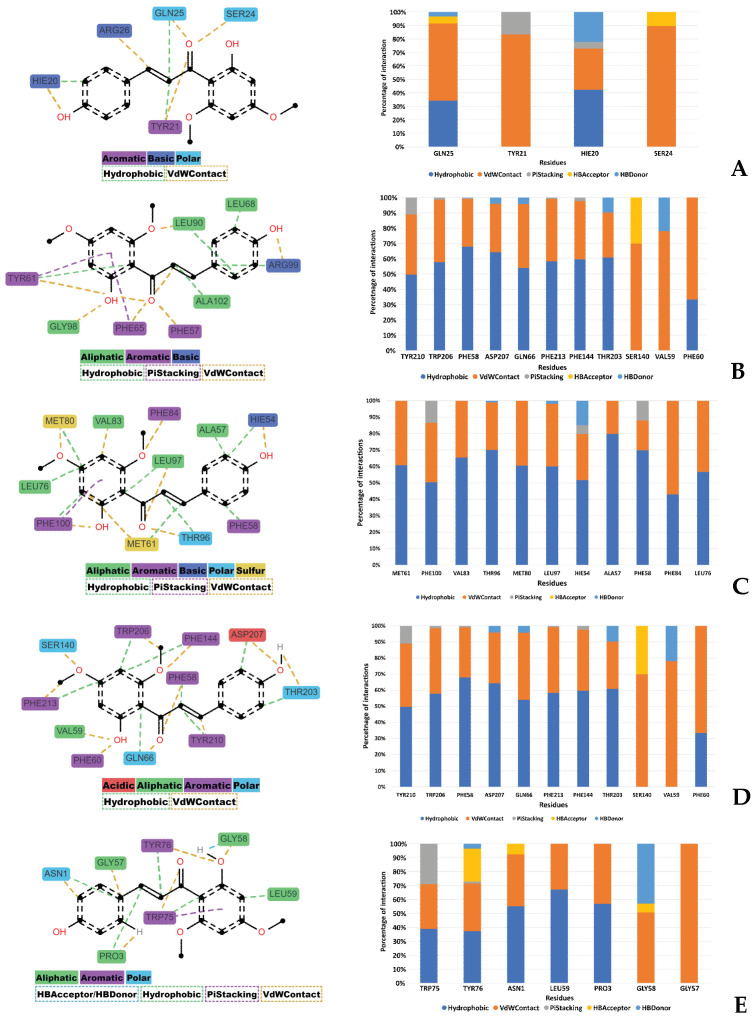
Depiction of the interactions between the proteins and the DDC ligand, emphasizing the residues that are engaged in interactions for more than 30% of the simulation’s duration. The subfigures represent different proteins: (**A**) Bcl-2, (**B**) Bcl-XL, (**C**) Mcl-1, (**D**) mTOR, and (**E**) XIAP.

**Figure 12 plants-13-00570-f012:**
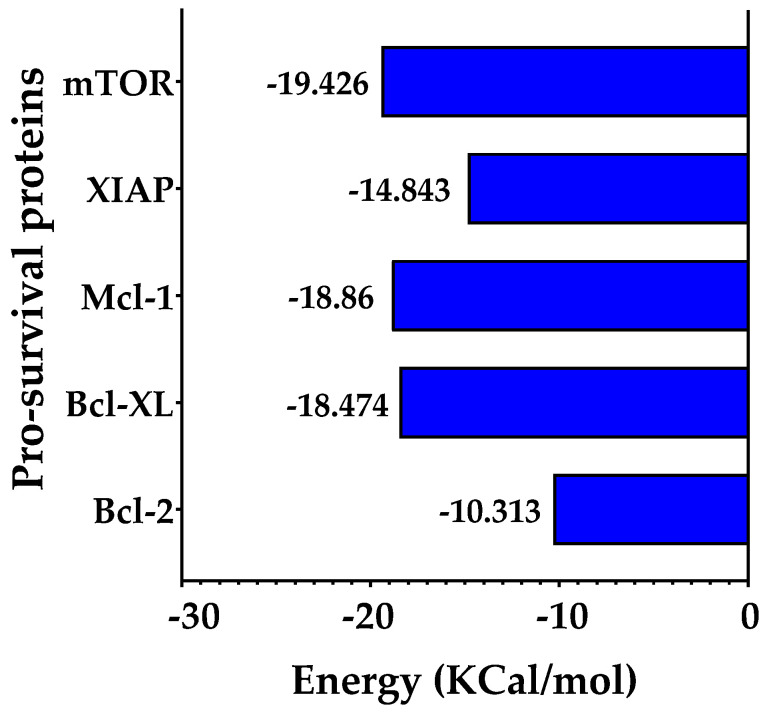
Binding free energy values calculated using MM-PBSA for each protein in complex with the ligand DDC.

**Table 1 plants-13-00570-t001:** ^1^H and ^13^C NMR (300 MHz, Acetone-d_6_) and UV nm spectral data for 2′,4-dihydroxy-4′,6′-dimethoxy-chalcone (DDC) obtained from the inflorescences of *Ch. tacotana*.

2′,4-Dihydroxy-4′,6′-Dimethoxy-Chalcone (DDC)						
Position	NMR		Displacement Analysis			
	δH (J in Hz)	δC (ppm)				
C-1	-	127.1	Reactive	Band I (nm)	Band II (nm)	Displacement (nm)
C-2	7.64 _(2H, dd, J = 2.4 Hz)_	130.5				
C-3	6.94 _(2H, dd, J = 2.4 Hz)_	115.93	MeOH	367	240	-
C-4	-	159.83				
C-5	6.94 _(2H, dd, J = 2.4 Hz)_	115.93	MeOH + MeONa	402	245	B II:5, B I:35
C-6	7.64 _(2H, dd, J = 2.4 Hz)_	130.5				
C-α	7.78 _(1H, d, J = 15.5 Hz)_	124.15				
C-β	7.91 _(1H, d, J = 15.5 Hz)_	142.84	MeOH + AcONa	372	245	B II:5, B I:5
C-β′	-	192.47				
C-1′	-	105.92	MeOH + AcONa + H_3_BO_3_	372	240	B I:5
C-2′	-	166.38				
C-3′	6.10 _(2H, dd, J = 2.4 Hz)_	93.77				
C-4′	-	168.26	MeOH + AlCl_3_	383	240	B I: 16
C-4′-OCH_3_	3.88 _(3H, s)_	55.17				
C-5′	6.10 _(2H, dd, J = 2.4 Hz)_	90.87				
C-6′	-	162.79	MeOH + AlCl_3_ + HCl	476	240	B I:9
C-6′-OCH_3_	4.02 _(3H, s)_	55.60				

## Data Availability

The data presented in this study supporting the results are available in the main text and Appendix A. Additional data are available upon reasonable request from the corresponding author. The data are not publicly available due to [according to the contracts signed with the Ministry of Environment and Sustainable Development, specifically the contract for access to genetic resources and their derived products, in accordance with the provisions of Decrees 730 of 1997 and 3570 of 2011, it is not possible to release the information on Colombian edemic species, except for prior pertinent procedures for their release.].

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
