# Peer review of "Natural 2′,4-Dihydroxy-4′,6′-dimethoxy Chalcone Isolated from Chromolaena tacotana Inhibits Breast Cancer Cell Growth through Autophagy and Mitochondrial Apoptosis"

_plants, 2024, doi:10.3390/plants13050570_

Round 1

Reviewer 1 Report

Comments and Suggestions for Authors

The present study “Natural 2',4-dihydroxy-4',6'-dimethoxy chalcone isolated from Chromolaena tacotana inhibits breast cancer cell growth through autophagy and mitochondrial apoptosis” show the potential anticancer properties of the abovementioned chalcone isolated from the inflorescences of the plant. The manuscript is clear and well-structured and presents interesting results. Nevertheless, it still needs some points to be addressed to improve its readability and impact. 

The introduction is informative; however, authors could be improving the Chromolaena tacotana description including previous results about its anticancer activity.  

The experimental design of the presented work is clear and several important aspect of cell death mechanisms (autophagy, apoptosis) are presented by various assays. Data presented both for normal and cancer cell types. In silico methods are also showed interaction of the chalcone with several anticancer proteins

Please provide producers of used equipment, reagents

The conclusions are appropriate; however, they could focus more on the potential of the plant and its perspective in this anticancer activity.

Author Response

We appreciate your feedback. We have adjusted the introduction by including previous results on the anticancer activity of Chromolaena tacotana, as well as in lines 123-131. The conclusions now focus more on the potential of the plant and its perspective in this anticancer activity.

All commercial producers of equipment and reagents were included.

Reviewer 2 Report

Comments and Suggestions for Authors

Manuscript entitled- Natural 2',4-dihydroxy-4',6'-dimethoxy chalcone isolated from Chromolaena tacotana Inhibits Breast Cancer Cell Growth Through Autophagy and Mitochondrial Apoptosis.

I have no technical comments this time, as the manuscript is very well written. 

I will give a straight acceptance. 

Author Response

Thank you for your comments, we have reviewed the document again to improve it even further.

Reviewer 3 Report

Comments and Suggestions for Authors

The goal of this study is to find out how well a chalcone derivative from Ch. 26 tacotana inflorescences can cause Luminal A and triple-negative breast cancer cells to go through controlled autophagy and intrinsic apoptosis. The article is well written, and the data is shown in a clear manner. Therefore, it can be accepted for publication after changes.
1. Provide the chemical structure of natural 2',4-dihydroxy-4',6'-dimethoxy chalcone.
2. Discuss the selection of concentrations used in the studies.
3. Word such as anti-cancerigen (line number 60) is not correct. Please thoroughly check the article for English mistakes.
4. Cite the relevant articles on flavonoids by Farhan et al. (for example, doi: 10.3390/metabo13040481).
5. The number of references used for a research article is very high (72 in total). I believe it must be reduced to the maximum possible extent.

Comments on the Quality of English Language

The english language is acceptable requiring minor changes.

Author Response

We appreciate your comments and follow your recommendations by making the following adjustments

  1. Provide the chemical structure of natural 2',4-dihydroxy-4',6'-dimethoxy chalcone:

ANSWER: The chemical structure is included in Figure 1 (lines 183-195)

  1.  Discuss the selection of concentrations used in the studies.

ANSWER: The in vitro assays were carried out at the minimum inhibitory concentration of the chalcone to analyze the different stages of the cell death mechanisms involved. Positive controls such as Paclitaxel, Combretastatin or Resveratrol were also used at IC50.

  1.  Words such as anti-carcinogenic (line number 60) is not correct. Please thoroughly check the article for English mistakes.

ANSWER: English was checked in all the text; the mistakes were corrected (for example anticarcinogenic in line 60)

  1. Cite the relevant articles on flavonoids by Farhan et al. (for example, doi: 10.3390/metabo13040481).

ANSWER: Thank you for the recommendation we include this work in our discussion. [46] Farhan, M.; Rizvi, A.; Aatif, M.; Ahmad, A. Current Understanding of Flavonoids in Cancer Therapy and Prevention. Metabolites 202313, 481. https://doi.org/10.3390/metabo13040481

  1. The number of references used for a research article is very high (72 in total). I believe it must be reduced to the maximum possible extent.

ANSWER: After including Ref 46, we carefully reviewed the references and removed two. The remaining references are relevant to each aspect evaluated and the technique used.